# ScalingCache: Extreme Acceleration of DiTs through Difference Scaling and Dynamic Interval Caching

**Lihui Gu**[1*†]**, Jingbin He**[2†]**, Lianghao Su**[2]**, Kang He**[2]**, Wenxiao Wang**[1✉]**, Yuliang Liu**[2✉]
[1]School of Software Technology, Zhejiang University
[2]KlingAI Research

## Abstract

Diffusion Transformers (DiTs) have emerged as powerful generative models, but their iterative denoising structure and deep transformer blocks incur substantial computational overhead, limiting the accessibility and practical deployment of high-quality video generation. To address this bottleneck, we propose ScalingCache, a training-free acceleration framework specifically designed for DiTs. Scaling-Cache exploits the inherent redundancy in model representations by performing lightweight offline analysis on a small number of samples and dynamically reusing previously computed activations during inference, thereby avoiding full computation at certain denoising steps. Experimental results demonstrate that ScalingCache achieves significant acceleration in both image and video generation tasks while maintaining near-lossless generation quality. On widely used video generation models including Wan2.1 and HunyuanVideo, it achieves approximately $2.5\times$ acceleration with only $0.5\%$ drop in VBench scores; on FLUX, it achieves $3.1\times$ near-lossless acceleration, with human preference tests showing comparable quality to original outputs. Moreover, under similar acceleration ratios, ScalingCache outperforms prior state-of-the-art caching strategies, achieving a $45\%$ reduction in LPIPS for text-to-image generation and $20-30\%$ reduction for text-to-video generation, highlighting its superior fidelity preservation. Our code is available at https://github.com/KlingAIResearch/ScalingCache.

## 1 Introduction

Recent advances in visual generation have established Diffusion Transformers (DiTs) as the dominant paradigm(Peebles & Xie, 2023), achieving state-of-the-art performance in modeling complex spatiotemporal patterns. However, their iterative denoising process incurs substantial computational cost, with generating even a few seconds of video often requiring several minutes(Sun et al., 2024; Wan et al., 2025). This efficiency bottleneck motivates the development of effective lightweight acceleration strategies.

Acceleration methods for DiTs can be broadly categorized into training-based and non-training-based approaches. Training-based methods, such as distillation(Zhang et al., 2025b), require large-scale data and computation, whereas non-training-based methods—including feature caching, sparsification(Xi et al., 2025; Xia et al., 2025; Yang et al., 2025), and quantization(Shang et al., 2023; Li et al., 2025; Zhang et al., 2025a)—can accelerate inference without additional training. Among these, feature caching leverages the temporal similarity between adjacent steps to improve efficiency. While all feature caching strategies are inherently lossy, the approximation error of existing methods remains too significant for professional-grade video generation, which demands near-lossless quality and high fidelity. This necessitates the development of more efficient and less destructive caching mechanisms.

The fundamental challenge of feature caching revolves around two core questions: how to use the cache and when to use it. For the former, naive approach is to directly reuse cached features, but this method faces a critical limitation: as temporal distance increases, feature similarity decays rapidly,

---

∗ Work done during an internship at KlingAI Research.
† These authors contributed equally to this work.
✉Corresponding authors. Email: wenxiaowang@zju.edu.cn, liuyuliang@kuaishou.com

leading to significant divergence. For the latter, a common strategy is to recompute features at fixed intervals, yet this rigid approach lacks the flexibility needed to adapt to the model's dynamic behavior.

Building on these two fundamental issues, prior studies have proposed various solutions. However, these approaches either lack flexibility(Chen et al., 2024; Liu et al., 2025b; Zhao et al., 2025) or fail to fully leverage the output features of each block to reduce prediction errors(Zhou et al., 2025; Liu et al., 2025a). To address these limitations, we propose two complementary strategies. First, we develop a more efficient predictive paradigm that mitigates the limitations of relying solely on differential scaling, while avoiding the exponential growth of activation caches required by higher-order taylor expansions. Second, we design a dynamic caching strategy that adaptively adjusts computational intervals during the denoising process.

Although Taylorseer(Liu et al., 2025b) employs higher-order Taylor expansions for block-level feature prediction within each module, increasing the expansion order provides little improvement in final performance while significantly increasing the storage and read/write overhead of the caches. Moreover, relying solely on first-order differences is insufficient to capture dynamic feature variations. This limitation motivates us to explore more efficient and expressive prediction paradigms.

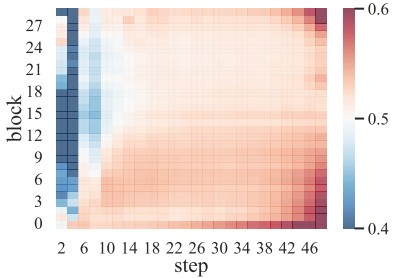

Figure 1: Lower values of $\sigma\big(\|\mathbf{y}^{(0)} - \mathbf{y}\|_1 - \|\mathbf{y}^{(1)} - \mathbf{y}\|_1\big)$ indicate that the current feature is more similar to the zero-order feature.

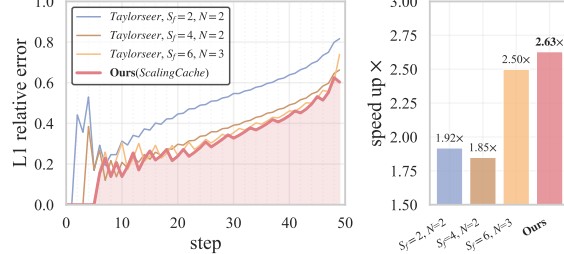

Figure 2: ScalingCache consistently achieves a higher acceleration ratio and lower L1 relative error compared to Taylorseer when evaluating the hidden state of the last block at each step under different caching strategies.

We observed that for certain transformer blocks at specific time steps, directly reusing cached pre-timestep features $\mathbf{y}^{(0)}$ yields smaller errors relative to full computation than applying first-order features $\mathbf{y}^{(1)}$ as shown in Figure 1. This finding suggests that combining the $\mathbf{y}^{(0)}$ and $\mathbf{y}^{(1)}$ is more effective than relying solely on either, motivating our introduction of differential scaling coefficients for each block are precomputed offline and applied during inference. During inference stage, we leverage precomputed differential scaling coefficient $\alpha$ for each time step and transformer block, enabling substantial improvements in generation quality.

The importance of different denoising steps varies significantly, making fixed cache intervals suboptimal. For instance, the initial full-computation steps $S_f$ are particularly critical: when $S_f$=2 and the full-computation interval $N$=2, the overall speedup is lower compared to the case where $S_f$=6 and $N$=3, yet the denoised output deviates more significantly from the original result as shown in Figure 2. Although several studies, including Teacache(Liu et al., 2025a) and Easycache(Zhou et al., 2025), have proposed dynamic caching strategies, these approaches typically base their policies only on the input to the first block and the output of the last block, without fully considering the dynamics of each intermediate block.

To this end, we propose **_ScalingCache_**, a caching framework that operates directly on hidden states of the transformer block to accelerate DiT inference. Our contributions are summarized as follows:

- Cache prediction with differential scaling optimization. We computed differential scaling coefficients for each time step and transformer block offline using a small set of samples, enabling accurate cache-based prediction.

- Runtime adaptive dynamic interval caching strategy. We propose an adaptive dynamic cache prediction approach that leverages both the outputs of each block and the variation of errors during computation, allowing the interval of full step computations to be adjusted adaptively.

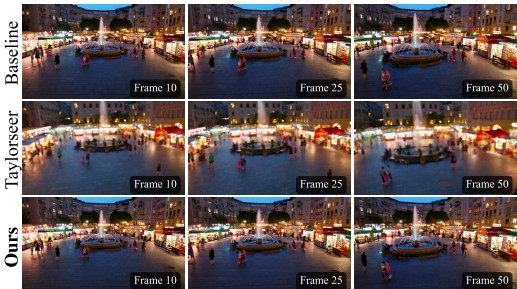 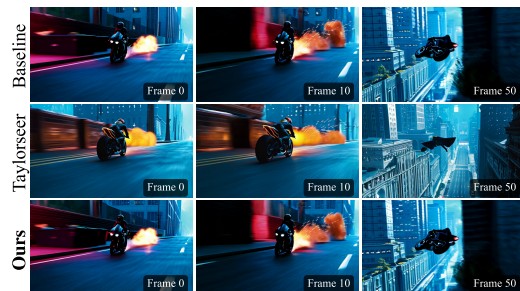

(a) Prompt: *The camera orbits around a bustling city square at dusk.* Taylorseer's scene generation is highly blurry, with a significant discrepancy compared to the original video.

(b) Prompt: *A sleek motorcycle shreds through neon-lit alleyways, bullets sparking off dumpsters.* Taylorseer's character is riding the motorcycle in reverse, and the motorcycle disappears at the moment it flies off.

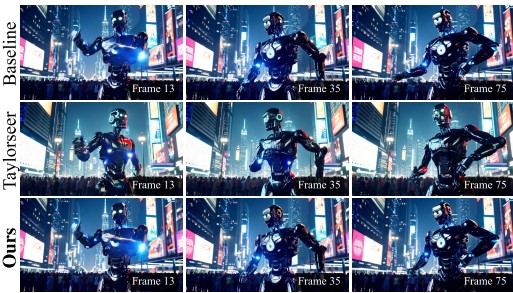 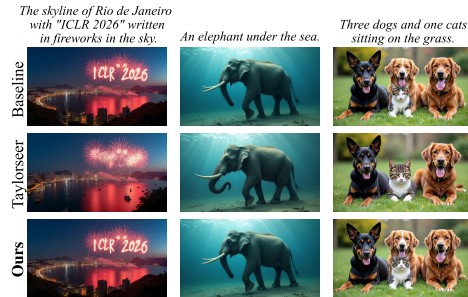

(c) Prompt: *CG animation digital art, a sleek and advanced robot standing in the bustling center of Times Square.* Taylorseer exhibited inconsistent lighting effects on the robot, with variations between the front and rear.

(d) In text-to-image generation task, Taylorseer(1.9×) failed to generate the "ICLR 2026" text in the first image, produced only one ivory tusk in the second image, and omitted the cat in the third image. In contrast, our method (2.2×) consistently ensured optimal results.

Figure 3: Video and image generation results on Wan2.1-14B and FLUX 1.dev. We highlight challenging scenarios where previous state-of-the-art methods (e.g., Taylorseer) produce noticeable artifacts or inconsistencies, while our method achieves nearly identical visual fidelity to the original videos/images even under a high acceleration factor.

- By synergistically combining the above modules for extreme acceleration, Scaling-Cache achieves significant speedup while maintaining near-lossless generation quality. Compared with prior state-of-the-art caching strategies, ScalingCache also demonstrates superior fidelity, achieving a 45% reduction in LPIPS for image generation tasks and a 20–30% reduction for video generation tasks.

## 2 RELATED WORK

Existing caching methods, including DeepCache (Ma et al., 2024b), Faster Diffusion (Li et al., 2023), and Cache-Me-if-You-Can (Wimbauer et al., 2024), are primarily designed for U-Net architectures. Their structural assumptions limit direct applicability to transformer-based DiTs, motivating dedicated caching mechanisms for this paradigm.

**Predictive Hidden-State Caching in Diffusion Transformers**. Caching approaches, such as AB Cache (Yu et al., 2025), PAB (Zhao et al., 2025), and TeaCache (Liu et al., 2025a), focus on directly reusing previously computed features without explicitly modeling their temporal evolution. While these methods reduce redundant computation, they often suffer from error accumulation and limited adaptability across different noise levels or generation conditions. $\Delta$-DiT (Chen et al., 2024) first introduced delta-based caching by incrementally updating attention and MLP activations across timesteps. The ToCa series (Zou et al., 2025; 2024) extend this idea by introducing dynamic feature correction, which alleviates iterative error accumulation during cache reuse. Building on this line, Taylorseer (Liu et al., 2025b) further enhances predictive caching by constructing cross-timestep

mappings that better preserve information flow in isotropic architectures. By leveraging the smooth continuity of hidden states across adjacent timesteps, these approaches achieve high efficiency without runtime scheduling, while offering stronger robustness than naive cache reuse.

**Dynamic scheduling caching.** Dynamic scheduling techniques adapt caching strategies at runtime by exploiting input characteristics or timestep patterns. Rule-based methods such as PAB (Zhao et al., 2025) adopt fixed-frequency attention reuse, gaining efficiency but lacking adaptability. Data-driven approaches improve flexibility: TeaCache (Liu et al., 2025a) fits polynomial mappings of timestep embeddings, while AdaCache (Kahatapitiya et al., 2024) performs online similarity checks to reuse block outputs, though both incur profiling or computation overhead. FORA (Selvaraju et al., 2024) reduces redundancy by selectively reusing spatio-temporal attention subsets, and later extensions such as L2C (Ma et al., 2024a) and ABCache (Yu et al., 2025) enable learnable layer selection or multi-step reuse. While dynamically responsive to runtime states, these methods face a common trade-off: heuristic rules risk quality degradation, whereas data-driven and adaptive schemes sacrifice efficiency due to profiling or similarity computation costs.

## 3 METHOD

### 3.1 OVERVIEW

Diffusion Transformers (DiTs) follow a hierarchical architecture, denoted as $\mathcal{M} = B_1 \circ B_2 \circ \cdots \circ B_L$, where each block $B_l$ consists of multiple distinct modules. For example, in Wan2.1 (Wan et al., 2025), each block comprises a cross-attention (CA) module conditioned on the time step and observations, a self-attention (SA) module, and a feed-forward network (FFN). This can be formally expressed as

$$B^l = \mathcal{F}_{SA}^l \circ \mathcal{F}_{CA}^l \circ \mathcal{F}_{MLP}^l, \quad l \in \{1, 2, \ldots, L\}, \tag{1}$$

where the superscript $l$ denotes the block index. Each module incorporates a residual connection, defined as $\boldsymbol{y}_t^l = \boldsymbol{x}_t^l + \mathrm{AdaLN} \circ f(\boldsymbol{x}_t^l)$, where $\mathrm{AdaLN}$ denotes adaptive layer normalization, and $f(\boldsymbol{x}_t^l)$ represents the function implemented by one of the modules within the block, i.e., $f \in \{\mathcal{F}_{SA}^l, \mathcal{F}_{CA}^l, \mathcal{F}_{MLP}^l\}$. Given an input $\boldsymbol{x}_t^l$ at step $t$, the output of the $l$-th block is denoted as $\boldsymbol{y}_t^l$.

### 3.2 DIFFERENTIAL SCALING FOR PREDICTION

In DiTs, each denoising step requires full computation of intermediate features. Naive feature reuse often neglects the dynamic evolution of features over time, potentially leading to the accumulation of approximation errors. To address this, Taylorseer proposes a linear prediction-based caching strategy. Its core idea is not only to cache feature values but also to record their temporal differences, enabling the prediction of features for future steps. Specifically, the first-order features at step $t$ can be predicted using the formula:

$$\boldsymbol{y}_t^{'l} = \boldsymbol{y}_\tau^l + \frac{k}{T}(\boldsymbol{y}_\tau^l - \boldsymbol{y}_{\tau-T}^l), \tag{2}$$

where $\tau = t - k$ denote the most recent full-computation step, and the second most recent full-computation step can be expressed as $\tau - T$ and $T$ represents the caching interval. The term $\Delta \boldsymbol{y}_\tau^l = (\boldsymbol{y}_\tau^l - \boldsymbol{y}_{\tau-T}^l)/T$ represents the average rate of change in the feature between these two time steps. This first-order prediction strategy effectively captures the linear temporal trend of feature evolution, significantly improving prediction accuracy compared to directly reusing cached features.

We observe that, for different denoising steps $t$ and different blocks $B_t^l$, the L1 error of first-order and zero-order features with respect to the full computation exhibits distinct regional patterns in Figure 1. This indicates that, for certain blocks $B_t^l$, directly reusing features can outperform first-order linear prediction. Motivated by this observation, we propose the following modified first-order linear prediction:

$$\hat{\boldsymbol{y}}_t^l = \boldsymbol{y}_\tau^l + \alpha_t^l k \Delta \boldsymbol{y}_\tau^l, \tag{3}$$

where $\alpha_t^l$ denotes the first-order differential scaling coefficients. In order to derive $\alpha_t^l$, we conduct an offline estimation on a collection of prompts. For each block $B_t^l$, we compute $\alpha_t^l$ by minimizing the discrepancy between the predicted and fully computed outputs via a least-squares formulation:

$$\min_{\alpha_t^l} ||\hat{\boldsymbol{y}}_t^l - \boldsymbol{y}_t^l|| = \min_{\alpha_t^l} ||\boldsymbol{y}_\tau^l - \boldsymbol{y}_t^l + \alpha_t^l k \Delta \boldsymbol{y}_\tau||. \tag{4}$$

During the practical phase of offline $\alpha_t^l$ estimation, we use $k$=1, $T$=1, the closed-form solution is given by:

$$\alpha_t^l = \frac{\langle \boldsymbol{y}_{t-1}^l - \boldsymbol{y}_t^l, -\Delta \boldsymbol{y}_{t-1}^l \rangle}{\langle \Delta \boldsymbol{y}_{t-1}^l, \Delta \boldsymbol{y}_{t-1}^l \rangle}, \tag{5}$$

where $\langle \cdot \rangle$ denotes the inner product. To improve stability and generalization, we update $\alpha_t^l$ using an exponential moving average $\alpha_t^l \leftarrow \beta \alpha'^l_t + (1 - \beta) \alpha_t^l$, where $\alpha'^l_t$ is a value obtained previously from a set of prompts. To better illustrate this mechanism, we provide a schematic overview in Figure 4. In practice, we precompute $\alpha_k^l$ for each block using approximately 50 prompts offline and $\beta$=0.97. This offline computation introduces no additional overhead during online inference.

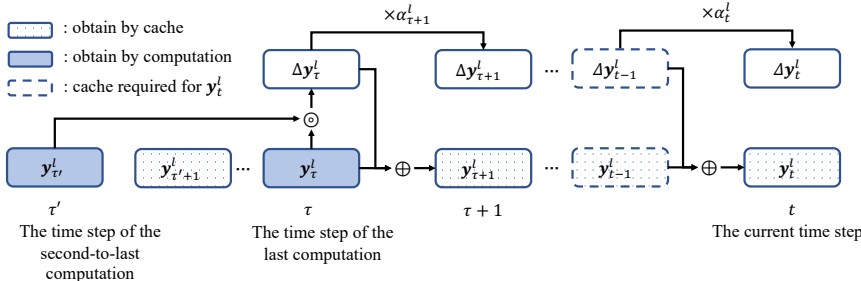

Figure 4: The complete process of obtaining current-step features via differential scaling-based block-level prediction.

The complete computation process for predicting the features at the current time step using single-stage differential scaling is illustrated in Figure 4. Note that each full computation updates $\Delta \boldsymbol{y}_\tau^l$. Since the scaling factors between two full computation steps is different, we use the following formula to obtain an estimate of $\Delta \boldsymbol{y}_\tau^l$:

$$\Delta \boldsymbol{y}_\tau^l = \boldsymbol{y}_\tau^l \circ \boldsymbol{y}_{\tau'}^l = \frac{(\boldsymbol{y}_\tau^l - \boldsymbol{y}_{\tau'}^l) \prod_{i=\tau'+1}^\tau \alpha_i}{\sum_{k=\tau'+1}^\tau \prod_{i=\tau'+1}^k \alpha_i^l}. \tag{6}$$

Our caching strategy requires storing two tensors per module: the cached feature $\boldsymbol{y}_{t-1}^l$ and the feature difference $\Delta \boldsymbol{y}_{t-1}^l$. In the Appendix G, we analyze the additional memory and computational overhead introduced by ScalingCache on various mainstream generative models.

### 3.3 RUNTIME DYNAMIC INTERVAL CACHING

In conventional feature caching strategies, full computation and cache updates are typically performed at fixed intervals or based on predetermined thresholds. However, our observations reveal a U-shaped error pattern in cache predictions: when optimized using first-order differences, intermediate timesteps exhibit relatively low prediction errors, whereas the beginning and end of the diffusion process show larger deviations. This indicates that static caching interval may incur unnecessary computational overhead or lead to the accumulation of approximation errors.

To address this issue, we propose a runtime dynamic interval caching strategy, which adaptively adjusts caching intervals to improve computational efficiency while maintaining prediction accuracy. For each timestep $t$, the dynamic error $e_t$ of step $t$ is defined as:

---

**Algorithm 1** ScalingCache inference strategy

---

**Input**: DiT model $\mathcal{M}$, $[\alpha_t^l]_{t=2,\ldots,N;l=1,\ldots,L}$
**Paramter**: $S_f$, initial warm-up steps
**Output**: $\{y_t^L|t=1,\ldots,N\}$, the output of the last block for each timestep

1: Initialize $\epsilon_t = 0, \mathcal{E} = \phi$
2: **for** $t = 1$ to $N$ **do**
3:     Calculate $\delta_s = 1/|\mathcal{E}|\sum_{\epsilon \in \mathcal{E}} \epsilon$
4:     **if** $t \in [0, S_f - 1]$ or $\epsilon_t > \delta_s$ **then**
5:         $y_t^L = \mathcal{M}(x_t)$
6:         $\epsilon_t = \bar{e}_t$
7:         $\mathcal{E} \leftarrow \mathcal{E} \cup \epsilon_t$
8:     **else**
9:         **for** $l = 1$ to $L$ **do**
10:           $y_t^l = y_{t-1}^l + \alpha_t^l \Delta y_{t-1}^l, \Delta y_t^l = \alpha_t^l \Delta y_{t-1}^l$
11:         **end for**
12:         $\epsilon_t = \epsilon_t + \bar{e}_t$, update cumulative error
13:     **end if**
14: **end for**

---

$$\bar{e}_t = \frac{1}{L}\sum_{l=1}^{L} \left\| \frac{y_t^l - y_{t-1}^l}{y_{t-1}^l} \right\|_1, \tag{7}$$

where $y_t^l$ denotes the output of the $l$-th block at timestep $t$, and $y_{t-1}^l$ corresponds to the last fully computed output. This metric quantifies the relative deviation between the predicted and fully computed features, providing a principled criterion for cache updates.

We further define the cumulative error from the last full computation to the current timestep as $\epsilon_t = \sum_{i=\tau}^{t-1} \bar{e}_t$. The cache update rule is then formulated as:

$$y_t^l = \begin{cases} f(x_t^l), f \in \{\mathcal{F}_{SA}^l, \mathcal{F}_{CA}^l, \mathcal{F}_{MLP}^l\}, & \text{if } \epsilon_t > \delta_s \text{ or } t \in [0, S_f - 1] \\ y_{t-1}^l + \alpha_t^l \Delta y_{t-1}^l, & \text{otherwise.} \end{cases} \tag{8}$$

where $S_f$ denotes the initial warm-up steps during which full computation is mandatory to capture the rapidly changing features of the early diffusion phase, and $\delta_s$ is a dynamic error threshold that bounds the deviation between predicted and fully computed features. Specifically, if the cumulative error exceeds $\delta_s$, a full computation is performed to prevent predictions from diverging significantly from the true features. Otherwise, the cache and predicted features are fused using an adaptive weight $\alpha_t^l$ to approximate the full computation output.

In practical scenarios, one key objective is to maximize the proportion of videos whose LPIPS falls below a specified threshold or whose PSNR exceeds a desired level, while maintaining the same acceleration ratio. We observe that as shown in Figure 5, for high-variation video generation tasks, a smaller threshold $\delta_s$ is required to achieve desirable results, whereas for slow-variation scenarios, a larger $\delta_s$ suffices and further yields higher speedup ratios. In practice, $\delta_s$ can be estimated from the first $S_f$ timesteps, enabling the model to assign appropriate thresholds for different types of video generation tasks. Under a fixed $S_f$ setting in ScalingCache, the computed $\delta_s$ for high-variation samples is intentionally lower than that for low-variation ones, thereby mitigating that high-variation samples require a lower $\delta_s$ to achieve visual quality comparable to that of low-variation samples.

To implement this adaptive caching mechanism, we design the inference procedure outlined in Algorithm 1. Specifically, the algorithm maintains a cumulative error set $\mathcal{E}$ to estimate the dynamic threshold $\delta_s$ during inference. For each timestep, the model decides whether to perform full computation or reuse cached features by comparing the current error $\epsilon_t$ against $\delta_s$. This enables the model to dynamically balance accuracy and efficiency, with aggressive reuse in stable regions and conservative updates in rapidly changing regions.

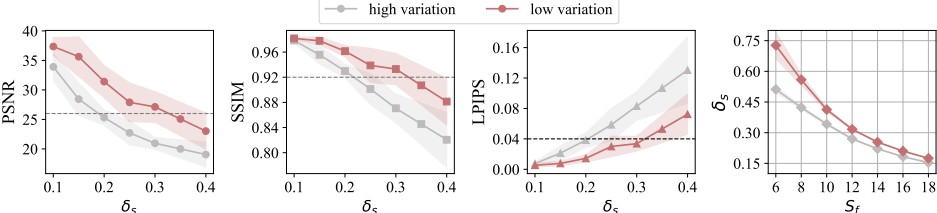

Figure 5: Employing two distinct sets of prompts—one with descriptive cues such as "still" (low variation) and another with "dramatically changing"(high variation)—we evaluate the generation quality under various threshold configurations using the Wan2.1 1.3B model.

We acknowledge that the current strategy may not be applicable to all instances—for example, cases that begin statically but later transition into dynamic states represent a known limitation of the present approach and warrant further investigation. Nevertheless, the method is effective in distinguishing the majority of samples and contributes to an overall improvement in performance.

# 4 EXPERIMENT

## 4.1 SETUPS

**Models.** We evaluate our proposed ScalingCache on text-to-video generation and extend the assessment to its generalization capability in text-to-image synthesis, with a focus on inference efficiency as well as generation quality. The experiments are conducted on three state-of-the-art visual generative models: the text-to-image generation model FLUX.1-dev(Labs, 2024), text-to-video generation model including Wan2.1-1.3B, Wan2.1-14B (Wan et al., 2025), and HunyuanVideo (Sun et al., 2024), to rigorously evaluate the acceleration and visual retention of ScalingCache.

**Evaluation Metrics.** For the primary task of text-to-video generation, we use default prompts in VBench (Huang et al., 2024) to assess visual retention. Specifically, we measure pixel-level fidelity, structural similarity, and perceptual consistency using PSNR, SSIM (Wang et al., 2004), and LPIPS (Zhang et al., 2018) against the original videos and images. We then systematically assess the generated results based on 16 core evaluation dimensions defined by the VBench framework to provide a comprehensive evaluation of the model's performance. For the text-to-image generation task, we perform inference on 200 DrawBench (Saharia et al.) prompts to generate images with a resolution of 1360×768. We then evaluate the generated samples using CLIP Score (Hessel et al., 2021) as key metrics to assess image quality and text alignment.

To capture subtle quality differences, especially in high-quality generated images, automated evaluation methods such as CLIP-score may not fully reflect these variations. To provide an objective assessment of the generated image quality, we employed a human preference-based comparison evaluation method. Each evaluator, given a specific prompt, was asked to select the image they considered superior or to judge if both images were of equal quality.

**Implementation Details.** We determine the alpha values using text prompts and examine the convergence of alpha with respect to the number of prompts, as illustrated in Figure 6. In the end, we use 20 prompts, and for each prompt, alpha is computed offline using five different random seeds. For the ablation studies on video generation, only one video is generated per prompt. For the large-scale evaluation on VBench, we use an NVIDIA H800 GPU to generate five video samples with different random seeds for each prompt, resulting in a total of 4,730 videos and 1,000 images are generated on DrawBench. Our method requires no parameter tuning and only involves specifying the value of $S_f$.

## 4.2 MAIN RESULTS

Table1 reports the performance of ScalingCache compared to several representative acceleration strategies on three text-to-video generation models, evaluated in terms of inference efficiency, visual retention and human preference evaluation.

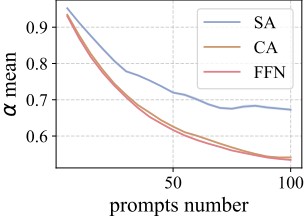

Figure 6: After performing 100 inference runs on Wan2.1 1.3B, the mean alpha values converge. For each number of inferences, we repeat the procedure five times and observe that the variance across runs is small.

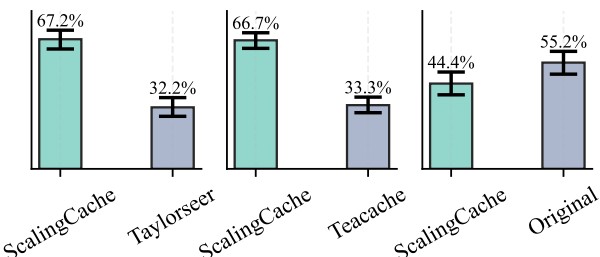

Figure 7: For the human preference evaluation, we compared the FLUX1.dev model's performance under a $3.1\times$ acceleration factor, evaluating our method, ScalingCache($S_f$=10), against Taylorseer ($S_f$=4, $T$=4) and ScalingCache against Teacache$_{0.6}$($2.0\times$).

Table 1: Comparison of ScalingCache with other acceleration methods on text-to-video tasks using the prompt-enhanced VBench dataset, reporting inference efficiency and visual quality metrics on representative models. Similar to Taylorseer, we quantify the reduction in computational complexity (measured by the decrease in FLOPs) to evaluate the theoretical speedup ratio.

| Methods | Efficiency | | Visual Retention | | | VBench (%) ↑ |
|---|---|---|---|---|---|---|
| | Latency (s) ↓ | Speedup ↑ | PSNR ↑ | SSIM ↑ | LPIPS ↓ | |
| Wan2.1 1.3B(Wan et al., 2025) (81 frames, 832×480) | | | | | | |
| Wan2.1 1.3B ($T = 50$) | 85.0 | 1× | - | - | - | 83.31 |
| + 40% steps | 34.1 | 2.5× | 14.50 | 0.523 | 0.437 | 80.30 |
| + Teacache$_{0.08}$ | 42.6 | 2.0× | 22.57 | 0.806 | 0.128 | 81.04 |
| + Taylorseer | 44.8 | 1.9× | 13.52 | 0.510 | 0.447 | 81.97 |
| + EasyCache | 34.2 | 2.5× | 25.24 | 0.834 | 0.095 | 82.48 |
| + **ScalingCache$_{10}$(ours)** | **34.0** | **2.5×** | **26.61** | **0.890** | **0.071** | **82.92** |
| Wan2.1 14B(Wan et al., 2025) (81 frames, 832×480), Ulysses×2, RingAttention×2 | | | | | | |
| Wan2.1 14B ($T = 50$) | 137.8 | 1× | - | - | - | 84.05 |
| + 50% steps | 68.9 | 2.0× | 15.82 | 0.696 | 0.336 | 79.36 |
| + TeaCache$_{0.14}$ | 91.9 | 1.5× | 18.60 | 0.688 | 0.244 | 83.95 |
| + MixCache | 81.1 | 1.8× | 23.45 | 0.814 | 0.124 | **83.97** |
| + **ScalingCache$_{10}$(ours)** | **55.1** | **2.5×** | **25.63** | **0.861** | **0.083** | 83.87 |
| HunyuanVideo (Sun et al., 2024) (129 frames, 960×544), Ulysses×2, RingAttention×2 | | | | | | |
| HunyuanVideo ($T = 50$) | 199.8 | 1× | - | - | - | 81.40 |
| + 50% steps | 100.1 | 2.0× | 17.57 | 0.734 | 0.247 | 78.78 |
| + TeaCache$_{0.1}$ | 133.7 | 1.5× | 23.85 | 0.819 | 0.173 | 80.87 |
| + MixCache | 110.5 | 1.8× | 26.86 | 0.906 | 0.060 | 80.98 |
| + Taylorseer | 72.2 | 2.8× | 26.57 | 0.860 | 0.135 | 80.74 |
| + EasyCache | 91.9 | 2.2× | 29.20 | 0.904 | 0.063 | 80.69 |
| + **ScalingCache$_{12}$(ours)** | **88.4** | **2.3×** | **30.80** | **0.930** | **0.049** | **81.13** |

**Inference efficiency.** ScalingCache consistently achieves the highest speedup across all evaluated models while maintaining low latency. For instance, on the Wan2.1 model, ScalingCache($S_f$=10) attains a $2.5\times$ speedup. A similar trend is observed on HunyuanVideo, where ScalingCache($S_f$=12) achieves a $2.3\times$ speedup, demonstrating its scalability across different model sizes and video lengths.

**Visual retention.** Despite aggressive acceleration, ScalingCache preserves superior visual quality. Across widely used video generation models including Wan2.1 and HunyuanVideo, Scaling-Cache achieves approximately $2.3$–$2.5\times$ acceleration with minimal impact on VBench scores (0.3–0.5% drop). On FLUX 1.dev, near-lossless $3.1\times$ acceleration is achieved, with all visual retention metrics significantly surpassing those of Taylorseer at $2.8\times$ acceleration. Under comparable accelera-

Table 2: Comparison of ScalingCache with other acceleration strategies on the FLUX 1.dev model.

| Methods | Efficiency | | Visual Retention | | | Clip Score (%) ↑ |
|---|---|---|---|---|---|---|
| | Latency (s) ↓ | Speedup ↑ | PSNR ↑ | SSIM ↑ | LPIPS ↓ | |
| FLUX 1.dev ($T = 50$) | 15.6 | 1× | - | - | - | **80.17** |
| + 50% steps | 7.8 | 2.0× | 29.36 | 0.683 | 0.318 | 78.88 |
| + TeaCache$_{0.6}$ | 7.8 | 2.0× | 28.08 | 0.400 | 0.690 | **81.79** |
| + Taylorseer$_3$ | 5.7 | 2.8× | 30.76 | 0.780 | 0.230 | 80.17 |
| + **ScalingCache$_{10}$(ours)** | **5.1** | **3.1×** | **32.28** | **0.819** | **0.131** | 80.25 |

tion ratios, ScalingCache consistently outperforms prior state-of-the-art caching methods, achieving a 45% reduction in LPIPS for image tasks and 20–30% reduction for video tasks, demonstrating its superior fidelity preservation.

**Human preference evaluation.** The accelerated images produced by ScalingCache were selected at a roughly equal rate as the original images, demonstrating that the accelerated generation preserves visual quality to a level comparable with the originals as shown in Figure 7.

The results demonstrate that ScalingCache effectively accelerates video and image generation with minimal quality loss. Its dynamic caching mechanism and predictive feature updates enable a superior trade-off between speed and fidelity, outperforming existing acceleration strategies across multiple models and video resolutions.

### 4.3 ABLATION STUDIES

Our ablation study on Flux1.dev and Wan2.1 1.3B demonstrates that both differential scaling coefficient($\alpha$) and dynamic caching intervals are critical for efficiency and generation quality. In the ablation setting without $\alpha$, we fix $\alpha$=1 and strictly set the static caching interval to the maximum value smaller than the dynamic caching interval divided by the acceleration factor. Under these conditions, visual fidelity drops significantly at higher acceleration factors. Introducing $\alpha$ mitigates feature prediction errors, improving PSNR, SSIM, and LPIPS with minimal overhead. Dynamic caching alone boosts acceleration while maintaining quality, and combining both strategies yields the best overall performance—substantially faster inference with negligible loss in visual quality. These results highlight that ScalingCache effectively allocates computational resources and preserves high-quality generation under high-acceleration settings as shown in Table 3.

ScalingCache only requires adjusting a single parameter, $S_f$, which can be tuned according to the desired acceleration. As shown in the Figure 8, for $S_f \leq 14$, all evaluated models achieve over 2.0× end-to-end inference speedup. In Figure 9, we explore the impact of higher acceleration ratios on the performance of Vbench by analyzing the changes in its individual sub-metrics.

Table 3: Ablation study on text-to-image and text-to-video task. We analyze the effect of $\alpha$ and dynamic caching on efficiency and visual retention.

| Model | Settings | | Speedup ↑ | Visual Retention | | |
|---|---|---|---|---|---|---|
| | $\alpha$ | dyn. | | PSNR ↑ | SSIM ↑ | LPIPS ↓ |
| Flux 1.dev | | | 2.9× | 29.15 | 0.652 | 0.324 |
| | ✓ | | 2.9× | 29.83 | 0.701 | 0.259 |
| | | ✓ | 2.6× | 31.04 | 0.772 | 0.192 |
| | ✓ | ✓ | **3.0×** | **32.28** | **0.819** | **0.131** |
| Wan2.1 1.3B | | | 2.4× | 24.53 | 0.857 | 0.092 |
| | ✓ | | 2.4× | 25.95 | 0.876 | 0.079 |
| | | ✓ | 2.5× | 22.50 | 0.809 | 0.129 |
| | ✓ | ✓ | 2.5× | **26.61** | **0.890** | **0.071** |

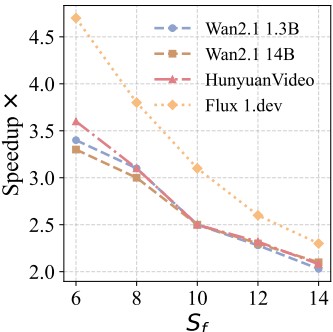

Figure 8: The speedup achieved across different models using various $S_f$.

Table 4: We analyze the stability of the $\alpha$ both within individual sub-tasks and across different sub-tasks for Wan2.1-1.3B. Each sub-task contains 10 prompts, and each prompt is generated 5 times using 5 different random seeds.

| **Sub-task** | motion | composition | human | material | mechanics | dynamic | static | random |
|---|---|---|---|---|---|---|---|---|
| $\|\alpha_i - \bar{\alpha}\|$ | 0.008 | 0.007 | 0.009 | 0.022 | 0.015 | 0.011 | 0.017 | **0.006** |

## 4.4 CROSS-TASK ROBUSTNESS ANALYSIS

Based on the VBench2(Zheng et al., 2025) dataset, we selects 5 representative sub-tasks as the evaluation foundation. To enhance the comprehensiveness of the assessment, we additionally construct two types custom-designed prompt sets—dynamic prompts and static prompts—resulting in a total of 8 distinct evaluation sub-tasks including a "random" sub-task. Each sub-task includes 10 carefully crafted text prompts to ensure diversity and representativeness across tasks. For each prompt, the alpha value is computed using five different random seeds. We further evaluate both the standard deviation of alpha values within each sub-task and the global standard deviation across all sub-tasks, so as to systematically analyze the stability of alpha computation.

As illustrated in the Table 4, we observe that the majority of subtasks yield alpha values within a deviation of 2.5% from the global mean, indicating that $\alpha$ demonstrates good cross-task stability. Since the $\alpha$ calculated from randomly sampled subsets show smaller deviations from the global $\bar{\alpha}$, we therefore recommend using a diverse set of prompts for $\alpha$ calculation in practical applications.

Table 5: Evaluation of the Wan2.1 1.3B model yielded two key findings: Firstly, using a custom alpha calculated per sample yields videos with the highest visual fidelity. Secondly, even in subtasks where the custom alpha deviates significantly from the mean alpha, using the mean alpha, while suboptimal, still produces results substantially superior to those generated without any alpha.

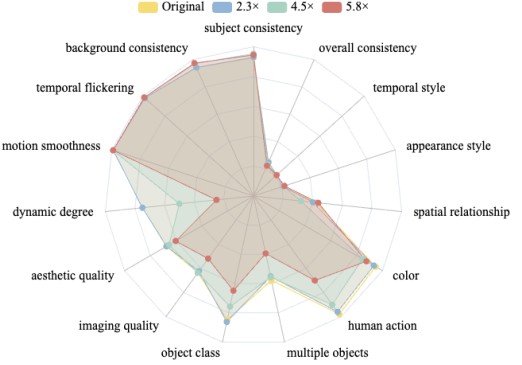

| Wan2.1 1.3B | Visual Retention | | |
|---|---|---|---|
| | PSNR↑ | SSIM↑ | LPIPS↓ |
| custom $\alpha$ | **27.76** | **0.939** | **0.034** |
| mean $\alpha$ | 26.92 | 0.935 | 0.037 |
| w/o $\alpha$ | 26.37 | 0.933 | 0.044 |

Figure 9: We vary $S_f$ to evaluate VBench and Visual Retention under higher acceleration. Reducing $S_f$ from 6 to 4 noticeably degrades generation quality. Although the overall VBench score drops by only 2%, higher acceleration significantly harms dynamic degree and human action performance.

## 5 CONCLUSION

In this study, we propose ScalingCache, an efficient inference acceleration framework for Diffusion Transformers. By introducing first-order differential scaling coefficients, the method significantly reduces computational overhead while maintaining negligible loss in generation quality. This optimization leverages temporal feature evolution trends through linear prediction, combined with a runtime dynamic caching mechanism that adaptively updates cached features based on cumulative error. Specifically, the differential scaling formulation enables lightweight yet accurate estimation of intermediate features, substantially decreasing the need for full network evaluations. Experiments show that it achieves 2.3–3.1× acceleration across text-to-video and text-to-image generation, with minimal impact on visual fidelity and strong performance in human preference evaluations. Compared with prior state-of-the-art caching methods, ScalingCache consistently delivers superior visual retention, highlighting its effectiveness and scalability for high-quality, resource-efficient generative inference.

ACKNOWLEDGEMENTS

This work was supported in part by The National Nature Science Foundation of China (Grant NO.: 62303406), in part by Ningbo Key R&D Program (NO.: 2025Z055), in part by Yongjiang Talent Introduction Programme (Grant NO.: 2023A-194-G).

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

# A    DECLARATION ON THE USE OF LLMs

In compliance with the conference policy on the use of Large Language Models (LLMs), we declare that LLMs were employed solely as an auxiliary tool for language polishing and for generating Python scripts dedicated to statistical analysis and visualization. The LLM contributed neither to research ideation, reasoning, nor any substantive aspect of the content. Therefore, no separate section has been included to describe LLM usage. The authors assume full responsibility for the entire manuscript, including all AI-assisted portions.

# B    DIFFERENTIAL SCALING COEFFICIENTS

The differential scaling coefficients remain consistent between the uncond stream and the cond stream, with most values falling within the range of 0.8 to 1.0, exceeding 1.2 are rarely observed.

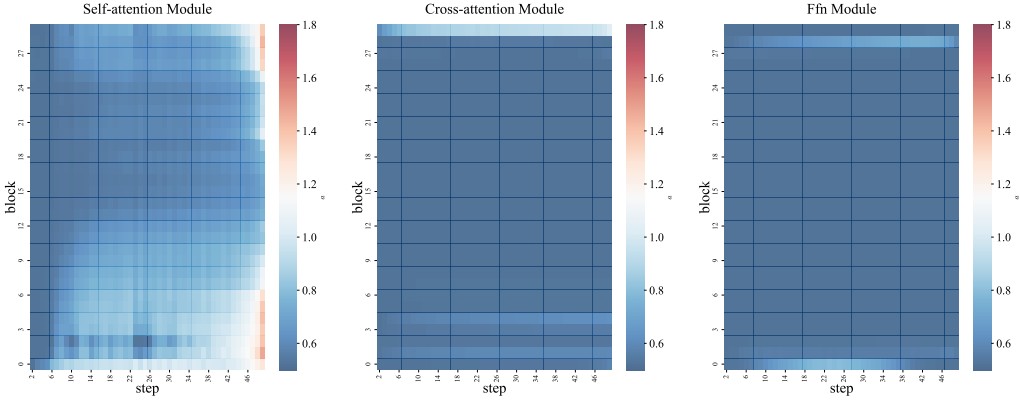

Figure 10: Wan2.1 1.3B conditional stream

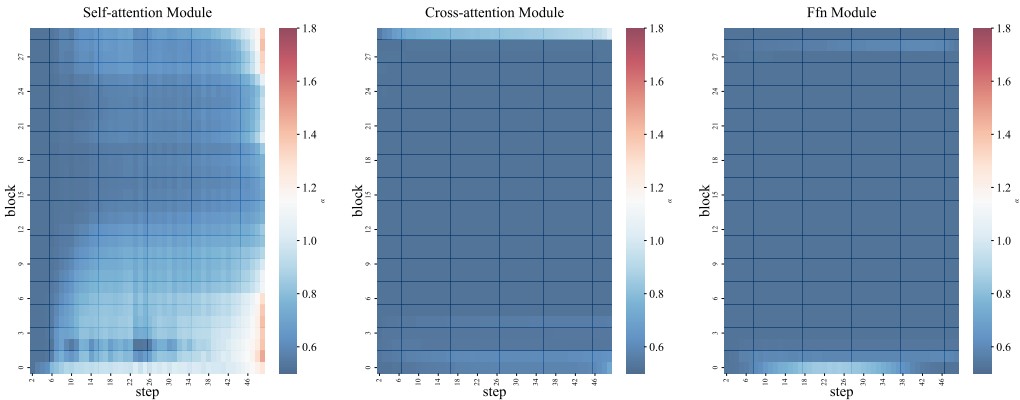

Figure 11: Wan2.1 1.3B unconditional stream

# C    IMAGE-TO-VIDEO RESULT

As shown in Table 6, our proposed method is evaluated on the Image-to-Video (i2v) generation task. The experimental results demonstrate that the Scalingcache technique is equally effective when applied to this task.

Table 6: Performance on the i2v task of Vbench2 using the ulyssess $\times$ 2 parallel configuration. Taylorseer uses an interval of 2 and enforces full computation for the first 4 steps and the last 2 steps.

| Methods | Efficiency | | Visual Retention | | |
|---|---|---|---|---|---|
| | Latency (s) $\downarrow$ | Speedup $\uparrow$ | PSNR $\uparrow$ | SSIM $\uparrow$ | LPIPS $\downarrow$ |
| Wan2.2 5B ($T = 50$) | 234.0 | 1$\times$ | - | - | - |
| + Taylorseer | 137.6 | 1.7$\times$ | 29.51 | 0.920 | 0.047 |
| **+ ScalingCache$_{10}$(ours)** | **111.4** | **2.1$\times$** | **33.29** | **0.959** | **0.021** |

## D   TEXT-TO-IMAGE CASE STUDY

A comparison between ScalingCache and Taylorseer on text-to-image generation tasks. The results demonstrate a clear advantage of ScalingCache, although some failure cases remain (e.g., the third row) in Figure 12. In most scenarios, however, the output is visually indistinguishable from the original, achieving near-lossless generation performance.

## E   WEIGHTED DIFFERENCE ACCUMULATION — DETAILED DERIVATION

We aim to estimate the hidden feature at time step $\tau$ in layer $l$ based on the previously cached feature at an earlier step $\tau'$ ($\tau' < \tau$). The forward dynamics between these steps can be expressed as a cumulative sum of residual updates:

$$y_\tau^l = y_{\tau'}^l + \sum_{k=\tau'+1}^{\tau} \Delta y_k^l, \tag{9}$$

where $y_k^l$ denotes the hidden feature at step $k$, and $\Delta y_k^l = y_k^l - y_{k-1}^l$ represents the residual change.

### E.1   EXPONENTIAL RESIDUAL DECAY ASSUMPTION

In practice, the magnitude of residuals tends to decrease as the sequence progresses, because earlier steps capture more global information and later steps mainly refine details. We model this behavior by assuming the residual norms decay approximately following a multiplicative factor $\alpha_k^l \in (0, 2)$:

$$\|\Delta y_{k+1}^l\| \approx \alpha_{k+1}^l \|\Delta y_k^l\|. \tag{10}$$

This implies that the residuals $\{\Delta y_k^l\}_{k=\tau'+1}^{\tau}$ form a geometric sequence scaled by $\alpha_i^l$.

### E.2   RELATION TO THE OBSERVED DIFFERENCE

The difference between the two hidden states is equal to the sum of their intermediate residuals:

$$y_\tau^l - y_{\tau'}^l = \sum_{k=\tau'+1}^{\tau} \Delta y_k^l. \tag{11}$$

If we denote $\Delta y_\tau^l$ as the most recent residual and back-propagate its magnitude along the sequence using the decay factors, each earlier residual can be written as:

$$\Delta y_k^l \approx \frac{\prod_{i=k+1}^{\tau} \alpha_i^l}{\prod_{i=\tau'+1}^{\tau} \alpha_i^l} \Delta y_\tau^l. \tag{12}$$

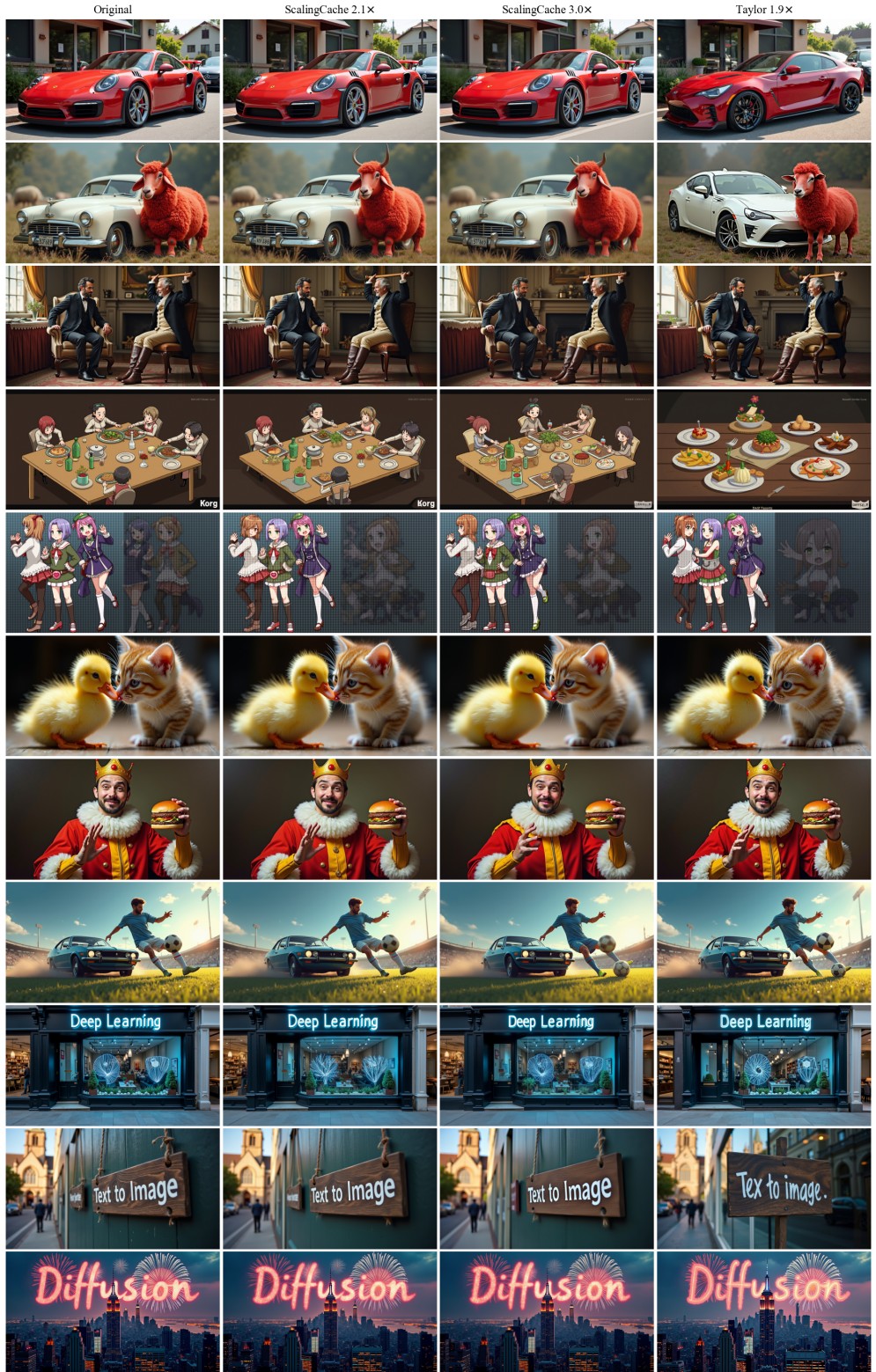

Figure 12: A comparison between ScalingCache and Taylorseer on the Flux1.dev model for the text-to-image task and by adjusting the scaling factor $S_f$ to 10 and 14, we obtain different speedups of $\times 3.0$ and $\times 2.1$, respectively.

### E.3 SOLVING FOR THE LAST-STEP RESIDUAL

Plugging this relation into the telescoping sum constraint gives:

$$\boldsymbol{y}_\tau^l - \boldsymbol{y}_{\tau'}^l = \sum_{k=\tau'+1}^{\tau} \frac{\prod_{i=k+1}^{\tau} \alpha_i^l}{\prod_{i=\tau'+1}^{\tau} \alpha_i^l} \, \Delta\boldsymbol{y}_\tau^l. \tag{13}$$

Rearranging yields a closed-form estimate of the most recent residual:

$$\Delta\boldsymbol{y}_\tau^l = \frac{(\boldsymbol{y}_\tau^l - \boldsymbol{y}_{\tau'}^l) \prod_{i=\tau'+1}^{\tau} \alpha_i^l}{\sum_{k=\tau'+1}^{\tau} \prod_{i=\tau'+1}^{k} \alpha_i^l}. \tag{14}$$

### E.4 USAGE IN DYNAMIC CACHE UPDATES

This formulation provides a stable way to approximate the last-step residual from two known hidden states $(\boldsymbol{y}_{\tau'}^l, \boldsymbol{y}_\tau^l)$ and a set of estimated decay factors $\{\alpha_i^l\}$. It can be used to refine cached hidden states during dynamic cache updates without recomputing all intermediate steps, thus reducing computational overhead.

## F DISTRIBUTED AND PARALLEL COMPUTING INTEGRATION

To enable efficient computation under sequence-parallel DiT models (e.g., Ulysses or Ring Attention), we compute the dynamic error $\bar{e}_t$ at each timestep $t$ in a distributed manner across all participating devices. Concretely, for each device $d \in 1, \ldots, D$ and each transformer layer $l \in 1, \ldots, L$ within a stream, we first compute the **local relative change** between consecutive timesteps:

$$e_t^{(d,l)} = \left\| \frac{\boldsymbol{y}_t^{(d,l)} - \boldsymbol{y}_{t-1}^{(d,l)}}{\boldsymbol{y}_{t-1}^{(d,l)}} \right\|_1. \tag{15}$$

We then aggregate the maximal local error across all modules on the same device for each layer, and average over layers to obtain the device-level local error:

$$\tilde{e}_t^{(d)} = \frac{1}{L} \sum_{l=1}^{L} \max_{f \in \{\mathcal{F}_{SA}^l, \mathcal{F}_{CA}^l, \mathcal{F}_{MLP}^l\}} e_t^{(d,l,f)}. \tag{16}$$

Finally, we perform an all-reduce operation over all participating devices to obtain the global dynamic error $\bar{e}_t$:

$$\bar{e}_t = \frac{1}{D} \sum_{d=1}^{D} \tilde{e}_t^{(d)}, \tag{17}$$

where $D$ denotes the world size. This distributed reduction step ensures that $\bar{e}_t$ consistently reflects the average prediction dynamics over all devices in the parallel group, enabling our method to adapt caching intervals coherently in a sequence-parallel setting.

## G OVERHEAD OF SCALINGCACHE ANALYSIS

We provide a detailed analysis of the additional memory requirements and associated overheads introduced by the feature caching mechanism.

**Memory Overhead**. The memory footprint for caching intermediate features is substantial. Using the Wan2.1 14B model as a primary example, the model contains 40 layers ($l$), and 2 computational streams ($S=2$). For each layer, features of dimension $(B, L, D)$ are stored. Since the algorithm

Table 7: A comparative analysis of generation quality and inference speed under different $S_f$ for HunyuanVideo.

| $S_f$ | Speedup↑ | Visual Retention | | | Vbench(%) |
|---|---|---|---|---|---|
| | | PSNR↑ | SSIM↑ | LPIPS↓ | |
| 4 | 5.8× | 18.59 | 0.684 | 0.341 | 75.18 |
| 6 | 4.5× | 23.62 | 0.813 | 0.172 | 79.67 |
| 12 | 2.3× | 30.80 | 0.930 | 0.049 | 81.13 |

requires caching both the previous feature and the delta ($N$=2), the total caching memory can be calculated as:

$$B \times L \times D \times l \times S \times N \times 2 Bytes$$

using the following parameters:

- B=1, L=32760, D=5120
- Data type: BF16 (2 bytes per element)

Applying the same method to other models yields the following additional memory requirements:

- Wan2.1 1.3B: $\sim$11.7GB
- Wan2.1 14B: $\sim$50GB
- Hunyuan Video: $\sim$62.2GB

The substantial memory overhead can be effectively addressed with sequence parallelism methods like Ring Attention or Ulysses. In these approaches, each attention head processes only a local sequence segment, meaning only the corresponding feature segments must be cached per device. This distributes the caching load evenly across the GPU cluster. For example, using 8-way sequence parallelism with the Wan2.1 14B model reduces the additional memory requirement to under 9 GB per GPU.

**Computational Overhead**.The core operations during the cache step are element-wise, making them memory-bound. Execution time can therefore be estimated based on the GPU's memory bandwidth. For an NVIDIA H800 with a memory bandwidth of approximately 3.3 TB/s, the estimated time per cache step for the Wan2.1 14B model is around 0.05 seconds.

**Communication Overhead**. The communication overhead is negligible. At the end of each cache step, synchronization is only required for a small scalar statistic computed per device. With Pdevices, the aggregated data size for communication is merely about $2 \times P$ bytes per step, resulting in a minimal communication cost.

