# OpenReview forum: "ScalingCache: Extreme Acceleration of DiTs through Difference Scaling and Dynamic Interval Caching"
_ICLR.cc/2026/Conference — ICLR 2026 Poster_

### Official Review · Reviewer_BDzQ · 2025-10-24

**Soundness:** 3
**Presentation:** 2
**Contribution:** 2
**Rating:** 4
**Confidence:** 4

**Summary:**

This paper presents ScalingCache, a training-free acceleration framework for Diffusion Transformers (DiTs), focused on visual generative models for both image and video generation. The approach introduces an adaptive dynamic caching mechanism, leveraging block-wise differential scaling coefficients (precomputed offline) and runtime error-adaptive cache interval selection to reduce redundant computation during denoising inference. Experimental results on multiple state-of-the-art image and video DiTs show substantial speedup (up to 3.1×) with minimal loss in visual quality, outperforming previously established caching baselines in both fidelity and efficiency on several standard benchmarks.

**Strengths:**

1. The paper addresses a pressing bottleneck in generative modeling—accelerating the slow inference of DiTs—through an approach requiring no retraining.
2. The method section provides a systematic derivation of the prediction formula (Eq. for $\hat{\boldsymbol{y}}{t}^{l}$, Page 4), the estimation of scaling coefficients (Eq. for $\alpha{t}^{l}$), and the adaptive error-based update rule. Full derivations and algorithmic steps are offered in the main text and appendices.
3. Results across major models (Wan2.1, HunyuanVideo, FLUX) and diverse settings show consistently strong speedups and lower LPIPS/SSIM drops versus all baselines.

**Weaknesses:**

1. Several directly related and competitive caching acceleration methods—TokenCache (Lou et al., 2024) [1], Gradient-Optimized Cache (Qiu et al., 2025) [2], FastCache (Liu et al., 2025) [3], SpeCa (Liu et al., 2024) [4], DiTFastAttn (Kim et al., 2024) [5], and Dynamic Diffusion Transformer (Wang et al., 2024) [6]—are not cited, discussed, or included as baselines. While the paper does compare to prominent recent caching strategies, this omission leaves a gap in situating ScalingCache's novelty and advancement versus the current best practices.
2. The derivation of the dynamic error threshold $\delta_s$ in Algorithm 1 (Page 6) could benefit from further theoretical and empirical justification—currently, it is set based on an empirical mean of prior errors, and may be sensitive to outliers or sample diversity. The implications for worst-case quality loss (e.g., video flicker) as a function of $\delta_s$ are left unexplored.
3. While near-lossless acceleration is highlighted, the methodology is not extensively challenged at higher speedup factors, nor is there a rigorous exploration of when the trade-off between speed and quality breaks down (e.g., in especially long sequences, rare prompts, or highly dynamic scenes).
4. In some equations, notation such as $\Delta \boldsymbol{y}_{\tau}^{l}$ is used before being defined, and formatting is at times inconsistent (e.g., parameter lists in Algorithm 1). In Section 3.2 (Page 4), certain variables and indices are introduced abruptly, which may cause confusion for readers less familiar with blockwise DiT architectures.

[1] Lou J, Luo W, Liu Y, et al. Token caching for diffusion transformer acceleration[J]. arXiv preprint arXiv:2409.18523, 2024.

[2] Qiu J, Liu L, Wang S, et al. Accelerating diffusion transformer via gradient-optimized cache[J]. arXiv preprint arXiv:2503.05156, 2025.

[3] Liu D, Yu Y, Zhang J, et al. Fastcache: Fast caching for diffusion transformer through learnable linear approximation[J]. arXiv preprint arXiv:2505.20353, 2025.

[4] Liu J, Zou C, Lyu Y, et al. Speca: Accelerating diffusion transformers with speculative feature caching[J]. arXiv preprint arXiv:2509.11628, 2025.

[5] Yuan Z, Zhang H, Pu L, et al. Ditfastattn: Attention compression for diffusion transformer models[J]. Advances in Neural Information Processing Systems, 2024, 37: 1196-1219.

[6] Zhao W, Han Y, Tang J, et al. Dynamic diffusion transformer[J]. arXiv preprint arXiv:2410.03456, 2024.

**Questions:**

1. Can the authors provide a more systematic analysis of scaling/failure scenarios? For instance, under what prompt or video conditions does the method’s error substantially increase, and what diagnostic measures could be advised in practice?
2. How sensitive is the method to the quality and diversity of prompts used for offline estimation of $\alpha$ coefficients? Figure 6 shows convergence, but quantitative analysis across tasks would clarify real-world robustness.
3. Given the omission of several related works (see above), how does ScalingCache’s performance and computational overhead compare to FastCache, TokenCache, and GOC, both qualitatively and in speedup/fidelity metrics?

---

> ### Author Response · Authors · 2025-11-21
>
> We sincerely thank Reviewer BDzQ for the careful review and valuable comments. We are pleased that our work was recognized for providing a systematic derivation of the prediction formula. Here are our responses to the questions raised.​
>
> > Q1:  provide a more systematic analysis of scaling/failure scenarios.
>
> We sincerely thank the reviewer for raising this important and insightful point. We constructed a comprehensive set of subtasks derived from the VBench2 benchmark, further enhanced by manually curated dynamic and static samples, as well as randomly sampled instances. Each sub-task was designed with 10 diverse textual prompts​ and evaluated under 5 distinct random seeds​.
>
> For each individual sub-task $i$, we computed the corresponding metric $\alpha_i$. Additionally, we aggregated results across all samples​ to derive a global metric $\bar \alpha$. By analyzing the discrepancy between $\alpha_i$ and $\bar \alpha$, we quantitatively assessed the degree of deviation​ of each sub-task from the overall performance. This approach allows us to systematically evaluate task-specific variations and validate the generalizability of $\alpha$.
>
> | sub-task category      | $\lvert \alpha_i − \bar \alpha \rvert $ |
> |------------------------|----------------|
> | Motion-Rationality     | 0.008       |
> | Composition            | 0.007       |
> | Human-Interaction      | 0.009       |
> | Material               | 0.021       |
> | Mechanics              | 0.014       |
> | Dynamic                | 0.011       |
> | Static                 | 0.017       |
> | Random                 | **0.006**       |
>
> - **Potential Failure Case 1**: For the "material"​ and "static"​ subtasks, the observed deviations are relatively large. This suggests that these subtasks alone may not reliably estimate $\alpha$ when used in isolation, as their metrics diverge significantly from the global $\alpha$.
> - **Potential Failure Case 2**: Individual samples within sub-tasks can still exhibit substantial deviations​ from the global $\bar \alpha$.
>
> To investigate case 2 further, we identified the top 30% of prompts​ with the largest absolute differences​ between their individual.
>
> | Method          | PSNR$\uparrow$  | SSIM$\uparrow$   | LPIPS$\downarrow$  |
> |-----------------|--------|-------|--------|
> | sample-specific $ \alpha$    | 27.76  | 0.939 | 0.034  |
> | mean $\alpha$      | 26.92  | 0.935 | 0.037  |
> | w/o $\alpha$       | 26.37  | 0.933 | 0.044  |
>
> The experimental results demonstrate that while the mean $\alpha$​ approach does not perform as well as using sample-specific $\alpha$ , it still significantly outperforms the baseline method without $\alpha$.
>
> > Q1: What diagnostic measures could be advised in practice?
>
> Here are our practical recommendations:
>
> - Since $\alpha$ calculated from randomly sampled show smaller deviations from the global $\bar \alpha$, we recommend using more diverse prompts for $\alpha$ computation to obtain more stable and reliable $\alpha$ estimates.
> - We suggest establishing a dedicated prompt test set, with particular emphasis on including additional prompts for underperforming sub-tasks. This targeted approach would help improving $\alpha$ estimation for problematic scenarios.
>
> > Q2: Quantitative analysis across tasks would clarify real-world robustness.
>
> We thank the reviewer for this insightful comment. We have added a discussion on this aspect in Section 4.4 of the paper. As shown in Table above, the mean value of the global $\alpha$ is 0.58. We found that the alpha values calculated from different majority subtasks deviate by no more than 2.5%, indicating that $\alpha$ exhibits good cross-task stability.
>
> > Q3: The performance and computational overhead of ScalingCache are compared to those of more prior works.
>
> These works are primarily conducted on image generation using the DiT/XL model. We  incorporate ScalingCache adaptations specifically for DiT/XL and conducted corresponding experimental analyses.
>
>
> | Method              | TFLOPS   |   FID$\downarrow$ | sFID$\downarrow$   | Speedup   |
> |:--------------------|:---------|------:|:-------|:-----------|
> | DDIM-50 steps | 23.74    |  2.32 | 4.32 | 1.00$\times$|
> | FastCache   | /   |  4.46 | /      | 2.10$\times$|
> | TokenCache  | 14.45|  2.37 | 4.53   | 1.32$\times$ |
> | GOC + FORA   | 8.90 |  3.52 | 4.81   | 2.67$\times$|
> | FORA  | 8.90|  3.87 | 5.19   | 2.67$\times$|
> | Scalingcache($S_f$=10) | 14.21|  2.36 | 4.43   | 1.67$\times$|
> | Scalingcache($S_f$=6)  | 11.04 |  2.40  | 4.44   | 2.15$\times$|
> | Scalingcache($S_f$=4)  | 8.79|  2.65 | 4.85   | 2.70$\times$|
>
>
> We provide a comparison between ScalingCache and Speca under extreme acceleration settings on HunyuanVideo.
>
> | Method      | Speedup| PSNR$\uparrow$  | SSIM$\uparrow$   | LPIPS$\downarrow$  | Vbench(%)  |
> |-------------|--------|-------|--------|--------|---------|
> | Speca   | 4.2$\times$   | 23.33 | 0.7992 | 0.1824 | 79.91   |
> | ScalingCache| 4.5$\times$   | 23.62 | 0.8131 | 0.1720 | 79.67   |

---

> ### Author Response · Authors · 2025-11-21
>
> It should be noted that DitFastAttn, like SVG, is an attention acceleration method and does not involve feature caching acceleration. Dynamic Diffusion Transformer​ is an approach that adjusts the model width at specific timesteps, rather than a caching strategy. It explicitly states that this method can be compatible with efficient caching approaches​ to achieve further acceleration. Therefore, we did not include a comparison with these works.
>
> > W2: $\delta_s$ is relatively sensitive to outliers or sample diversity.
>
> In response to the reviewer's concerns, we have supplemented the experimental comparison between using a **dyanmic $\delta_s$** and using a **fixed $\delta_s$**. Considering the increase in the number of high-quality generated videos, we consider videos with LPIPS < 0.1 to be of relatively good quality. Therefore, our goal is to achieve a higher percentage of videos with LPIPS < 0.1 under the same acceleration ratio. The following statistics are provided:
>
> | Model           | SpeedUp | LPIPS  | dynamic $\delta_s$ | fixed $\delta_s$ |
> |----------------|---------|--------|-----------------------------|----------------------------|
> | Wan2.1 1.3B    | 2.5$\times$    | 0.071  | 82.68%   | 79.23%   |
> | Wan2.1 14B     | 2.5$\times$   | 0.083  | 69.49%   | 66.38%   |
> | HunyuanVideo   | 2.3$\times$    | 0.049  | 97.17% | 94.89%   |
>
> Although dynamically estimating $\delta_s$ brings overall benefits, there are still cases where certain abnormal samples fail. We define a failure sample as one where the LPIPS is calculated to be < 0.1 using the static threshold but > 0.1 when using the dynamic threshold. The proportion of such failure samples is extremely low — for Wan2.1 1.3B, it is only 0.3%.
>
> > W3: Is there a rigorous exploration of when the trade-off between speed and quality breaks down?
>
> We have provided the variation chart of Vbench sub-metrics for the HunyuanVideo model under high acceleration ratios in Section 4.3 of the paper, and have also included this table in Appendix Table 7.
>
> | Speedup| $S_f$ | Vbench(%) | PSNR $\uparrow$ | SSIM $\uparrow$  | LPIPS$\downarrow$  |
> |--------------------|-----|--------|-------|--------|--------|
> | 5.8$\times$      | 4   | 75.18  | 18.59 | 0.684  | 0.341  |
> | 4.5$\times$      | 6   | 79.67  | 23.62 | 0.8131 | 0.172 |
> | 2.3$\times$      | 12  | 81.13  | 30.80 | 0.930  | 0.049 |
>
> As the acceleration ratio increases, certain sub-metrics may drop sharply, such as "dynamic degree".
>
> ---
>
> Finally, we sincerely appreciate the reviewer's meticulous and insightful evaluation. The constructive feedback has been instrumental in refining our manuscript. The reviewer has perceptively identified several critical areas for improvement, including: (1) **the necessity for deeper cross-task stability analysis of the $\alpha$**; (2) **enhanced comparative analysis with state-of-the-art methodologies**; (3) **necessary corrections to mathematical formulations**; and (4) **the important direction of investigating ScalingCache's performance under additional acceleration scenarios**. We hope the additional experiments and related analyses can effectively address your concerns.

---

### Official Review · Reviewer_6bkW · 2025-10-28

**Soundness:** 3
**Presentation:** 3
**Contribution:** 3
**Rating:** 6
**Confidence:** 5

**Summary:**

The paper introduces ScalingCache, a training-free acceleration framework tailored for DiTs. By synergizing differential-scaling-based prediction with runtime-adaptive caching intervals, ScalingCache delivers significant speed-ups on both image and video generation while retaining near-lossless quality. Extensive experiments on Wan2.1, HunyuanVideo and FLUX show 2.3–3.1× acceleration with only 0.3–0.5 % VBench drop, and outperform prior state-of-the-art caching methods in LPIPS and other fidelity metrics, demonstrating superior fidelity-efficiency trade-offs.

**Strengths:**

1.	The proposed algorithm is clearly described, with a well-defined formulation and solid explanation.
2.	The manuscript is clearly structured and well-articulated, making it easy for readers to follow.

**Weaknesses:**

1.	The related work section overlooks the discussion of cache acceleration methods for UNet-based models, even though the cache acceleration technique for DiT-based models is an extension of and inspired by the earlier approaches developed for UNet-based models.
2.	It would be great if the proposed method could further improve the sampling speed of the distilled models.
3.	It's better to provide a user study to verify, through human evaluation, whether the generative performance of the method is close to the baseline.
4.	Is using 50 prompts sufficient to determine the appropriate scale?

**Questions:**

The authors are encouraged to further explore the applicability of the proposed approach to few-step distilled models.

---

> ### Author Response · Authors · 2025-11-21
>
> We sincerely appreciate the reviewer’s highly positive assessment of our work. We are particularly grateful for their kind comments regarding the manuscript’s clear structure and articulate presentation We respond to your concerns as follows.
>
> > W1: The related work section overlooks the discussion of cache acceleration methods for UNet-based models.
>
> We sincerely appreciate the reviewer's valuable suggestion regarding the UNet architecture's cache mechanism. **The following content has been incorporated into the revised manuscript:**
>
> "Although effective in reducing computational cost, existing caching strategies such as DeepCache[1] and Faster Diffusion[2] have been developed specifically for the U-Net architecture, leveraging its unique characteristics for feature reuse. Another approach, Cache-Me-if-You-Can[3], further incorporates teacher-student imitation to minimize caching artifacts. Given the high computational demands of the prevailing DiTs architecture, researchers are developing dedicated caching mechanisms for its transformer-based paradigm to address the challenge of transferring U-Net-oriented optimization methods."
>
> > W2: It would be great if the proposed method could further improve the sampling speed of the distilled models.
>
> Caching intermediate features to accelerate distilled models is a bold but worthwhile idea.​ To be honest, since prior work hasn’t provided experimental analyses on distilled models with fewer sampling steps, we didn’t experiment with them either. Distillation requires extra training—which isn’t always practical for industrial scenarios—whereas feature caching is a plug-and-play, zero-trainingsolution.
> If training resources are available, one would typically aim for one-shot distillation(e.g., extreme low-step settings like 5 steps or even fewer). But if our method can further speed up a distilled model… well, maybe that distillation just wasn’t pushed far enough to begin with.
>
> > W3: It's better to provide a user study to verify, through human evaluation.
>
> That's right! Our work in the paper has taken this into consideration. In particular, we found that image Reward scores, such as CLIP score [4] and ImageReward [5], are difficult to align with metrics that assess lower-level semantic losses, so it is still necessary to incorporate human evaluation. Our evaluation system is conducted by a professional QA team, based on rigorous evaluation standards in the field of image generation, ensuring objective results through multi-dimensional comparative analysis. However, since evaluating videos is significantly more costly, we opted to conduct visual comparison experiments with human evaluations on generated images.
>
> > W4: Is using 50 prompts sufficient to determine the appropriate scale?
>
> This is a very critical issue, as also pointed out by the other three reviewers.
> As shown in Figure 6 of the paper, \alpha gradually converges as the number of prompts increases. Increasing the number of prompts helps improve stability.
>
> Reviewers PyFd and BDzQ suggested further analyzing the robustness of alpha across tasks. We have added a discussion on this aspect in Section 4.4 of the paper. As shown in Table above, the mean value of the global $\bar \alpha$ is 0.58. We found that the $\alpha$ calculated from different majority subtasks deviate by no more than 2.5%, indicating that exhibits good cross-task stability.
>
> If you are interested, you may refer to our response to Reviewer BDzQ, where we have also included an analysis of cases where the alpha deviates significantly for specific samples.
>
> Based on the above findings, **we provide the following recommendations**:
>
> - Use more diverse prompts in practical applications.
> - It is recommended to construct a test prompt set and collect additional prompts for subtasks that show larger deviations.
>
> ---
> **Reference**
>
> [1] Ma, Xinyin, Gongfan Fang, and Xinchao Wang. "Deepcache: Accelerating diffusion models for free." Proceedings of the IEEE/CVF conference on computer vision and pattern recognition. 2024.
>
> [2] Li, Senmao, et al. "Faster diffusion: Rethinking the role of unet encoder in diffusion models." CoRR (2023).
>
> [3] Wimbauer, Felix, et al. "Cache me if you can: Accelerating diffusion models through block caching." Proceedings of the IEEE/CVF Conference on Computer Vision and Pattern Recognition. 2024.
>
> [4] CLIPScore: A Reference-free Evaluation Metric for Image Captioning, 2022.
>
> [5] ImageReward: Learning and Evaluating Human Preferences for Text-toImage Generation, 2023.
>
> ---
>
> Thanks for your recognition of the paper. We hope the above responses and analyses can address your concerns.

---

### Official Review · Reviewer_PyFd · 2025-10-30

**Soundness:** 2
**Presentation:** 2
**Contribution:** 3
**Rating:** 4
**Confidence:** 3

**Summary:**

The paper presents ScalingCache, a training-free inference acceleration framework specifically designed for Diffusion Transformers (DiTs), targeting image and video generation tasks. The core idea leverages the temporal redundancy in the hidden states during the denoising process in DiT. It conducts lightweight offline analysis on a small number of samples to precompute differential scaling factors and dynamically reuses previously computed activations during inference to bypass certain computation steps. This method achieves significant speedup while maintaining near-lossless generation quality, outperforming existing caching strategies. It particularly demonstrates better robustness in complex video generation scenarios.

**Strengths:**

1. Important and Practical Problem: The high computational cost of DiT significantly limits its deployment in real-world scenarios like video generation. The proposed training-free acceleration approach holds clear practical value.

2. Sophisticated Technical Design:
The introduction of the differential scaling factor α effectively combines zero-order and first-order predictions, addressing the issue of large prediction errors in certain layers seen with methods like Taylorseer.

3. Comprehensive Experiments with Outstanding Results:
Covers multiple SOTA models (Wan2.1-1.3B/14B, HunyuanVideo, FLUX);
Evaluates both image and video tasks with metrics including PSNR/SSIM/LPIPS/VBench/CLIP Score and human preference.

4. Engineering-Friendly: Only requires tuning one hyperparameter (Sf, i.e., number of initial full computation steps), without the need for training, fine-tuning, or complex scheduling logic, making it easy to integrate.

**Weaknesses:**

1. Generalization of α: The α coefficients need to be estimated offline using a small number of prompts (~50 prompts). While the paper claims convergence and low variance (Figure 6), it doesn’t verify its generalization to out-of-distribution prompts (e.g., extreme styles or rare objects). If α is sensitive to prompts, frequent re-estimation may be necessary for deployment.

2. (Section 3.3) The authors acknowledge that their strategy may fail for "static-to-dynamic" videos (e.g., a scene suddenly transitioning from stillness to high-speed movement). Such scenarios are not uncommon in real-world videos. Furthermore, as the denoising process can theoretically access tokens from all frames at every step, this raises the question of why such scenarios would significantly impact this strategy and whether a reasonable threshold can still be estimated.

3. While the authors claim "no additional inference overhead," the calculation of dynamic errors (Equation 7) and all-reduce operations (Appendix F) under sequence parallelism still involve communication and computational overhead.
The authors fail to report the storage cost of caching (storing y and Δy per module) and do not discuss the potential impact on devices with limited memory.

4. Lack of Comparisons: Why didn’t the authors compare their method with other acceleration approaches, such as Sparse VideoGen?

**Questions:**

1. In Table 1, why does MixCache achieve the highest score on the 14B WAN2.1 model? Could the authors explain this anomaly?

2. Concerns remain regarding the generalization of α. How does α perform in out-of-distribution prompts? If the selected prompts are not diverse enough, could this lead to suboptimal results?

3. Regarding the human evaluation experiments, were the participants professionals or anonymous general users? Could this introduce bias?

4. In the real-world deployment of 14B models, how much does the cache increase memory consumption?

---

> ### Author Response · Authors · 2025-11-21
>
> We sincerely appreciate Reviewer PyFd's careful review and professional comments, which have been immensely helpful in improving our work.
>
> > Q1: Why does MixCache achieve the highest score on the 14B WAN2.1 model?
>
> Thanks for carefully examining our experimental results. MixCache[1] is a contemporary approach that also aims for near-lossless feature caching acceleration. We reported the metrics provided in the MixCache paper, where the Vbench scores are all **relatively close to the original score of 84.05 — specifically, 83.97 (MixCache) and 83.87 (ScalingCache)**. Since MixCache has not been open-sourced, we are unable to conduct further analysis. However, ScalingCache still demonstrates clear advantages in terms of acceleration ratio and visual quality preservation (as measured by PSNR, SSIM, and LPIPS). **The Vbench score is also based on evaluations from a scoring model**, so minor differences like these should fall within the normal range when striving for near-lossless acceleration.
>
> Inspired by your comments, we found it valuable to conduct a more in-depth analysis of the results. We have added a comparison with the contemporary open-source work EasyCache[2], evaluating on the sub-metrics of Vbench as shown in Figure 13 of the paper. **We found that our advantage in Vbench scores largely stems from the "dynamic degree" dimension**, demonstrating our method's strength in handling dynamic scene changes.
>
> > Q2: Concerns remain regarding the generalization of $\alpha$.
>
> Thanks to the reviewer for pointing out the potential concern. Similarly, Reviewer 6bkW raised the same concern, and Reviewer BDzQ suggested further analyzing the robustness of alpha across tasks.
>
> We randomly selected a subset of sub-tasks from VBench2 and manually constructed dynamic and static scene tasks. Each task consists of 10 prompts and 5 random seeds.For each individual sub-task $i$, we computed the corresponding metric $\alpha_i$. Additionally, we aggregated results across all samples​ to derive a global metric $\bar \alpha$. By analyzing the discrepancy between $\alpha_i$ and $\bar \alpha$, we quantitatively assessed the degree of deviation​ of each sub-task from the overall performance. This approach allows us to systematically evaluate task-specific variations and validate the generalizability of $\alpha$.
>
> | sub-task category      | $\lvert \alpha_i − \bar \alpha \rvert $ |
> |------------------------|----------------|
> | Motion-Rationality     | 0.008       |
> | Composition            | 0.007       |
> | Human-Interaction      | 0.009       |
> | Material               | 0.021       |
> | Mechanics              | 0.014       |
> | Dynamic                | 0.011       |
> | Static                 | 0.017       |
> | Random                 | **0.006**   |
>
> As can be seen from the table,$\alpha$ exhibits a certain degree of generalizability across tasks. However, for the "material" and "static" subtasks, the observed deviations are relatively large. This suggests that these sub-tasks alone may not reliably estimate $\alpha$ when used in isolation, as their metrics diverge significantly from the global $\bar{\alpha}$.
>
> Based on the above findings, **we provide the following recommendations**:
>
> 1. Use more diverse prompts in practical applications.
> 2. It is recommended to construct a test prompt set and collect additional prompts for subtasks that show larger deviations.
>
> > W2: As the denoising process can theoretically access tokens from all frames at every step, this raises the question of why such scenarios would significantly impact this strategy and whether a reasonable threshold can still be estimated.
>
> As emphasized in the paper, the current strategy indeed cannot provide a reasonable estimate for cases where a video starts as static but later becomes dynamic. This is because the cached features $\mathbf{y}_{t}^{L} \in \mathbb{R}^{B \times (H \times W \times T) \times D}$ are used to compute $\bar{e}_t$ as a whole, which then leads to the derivation of $\delta_s$. **One feasible solution is to split the features along the temporal dimension $T$, compute $\bar{e}_t$ in blocks, and then derive $\delta_s$ through some form of global reduction strategy** — for example, by applying a max operation​ over the time dimension to capture the most significant deviation across frames.
>
> > Q3: About human evaluation experiments.
>
> Our evaluation system is conducted by a professional QA team, based on rigorous evaluation standards in the field of image generation, ensuring objective results through multi-dimensional comparative analysis.
>
>
> [1] Wei, Yuanxin, et al. "MixCache: Mixture-of-Cache for Video Diffusion Transformer Acceleration." arXiv preprint arXiv:2508.12691 (2025).
>
> [2] Zhou X, Liang D, Chen K, et al. Less is Enough: Training-Free Video Diffusion Acceleration via Runtime-Adaptive Caching[J]. arXiv preprint arXiv:2507.02860, 2025.

---

> ### Author Response · Authors · 2025-11-21
>
> > W4: Lack of Comparisons, such as SVG.
>
> It appears that the reviewers are also interested in related work on inference acceleration for video generation. Since SVG relies on sparse attention—a different approach to accelerating DiT-based generation (as opposed to feature caching)—we did not include a direct comparison with it in paper. Its reported acceleration results on HunyuanVideo (from the original paper) can still serve as a reference.
>
> Reviewer BDzQ also cited several related works. We have conducted comparisons with prior approaches on DiT/XL - please see my reply if you're interested.
>
> | Method       | SpeedUp   |   PSNR |   SSIM |   LPIPS |
> |:-------------|:----------|-------:|-------:|--------:|
> | SVG          | 1.4$\times$      |  26.57 |  0.859 |  0.1368 |
> | ScalingCache | 2.5$\times$     |  25.63 |  0.861 |  0.083  |
>
> > Q4: In the real-world deployment of 14B models, how much does the cache increase memory consumption?
>
> Thanksr for pointing out this critical issue. The memory footprint for caching intermediate features is substantial. Using the Wan2.1 14B model as a primary example, the model contains 40 layers ($l$), and 2 computational streams ($S$=2). For each layer, features of dimension ($B,L,D$)are stored. Since the algorithm requires caching both the previous feature and the delta ($N$=2), the total caching memory can be calculated as: $B\times L \times D \times l \times S \times N \times 2 Bytes$, using the following parameters:
>
> - $B$=1, $L$=32760, $D$=5120
> - Data type: BF16 (2 bytes per element)
>
> Applying the same method to other models yields the following additional memory requirements:
>
> - Wan2.1 1.3B: $\sim$11.7GB
> - Wan2.1 14B: $\sim$50GB
> - Hunyuan Video: $\sim$62.2GB
>
> The substantial memory overhead can be effectively addressed with sequence parallelism methods like Ring Attention or Ulysses. In these approaches, each attention head processes only a local sequence segment, meaning only the corresponding feature segments must be cached per device. **This distributes the caching load evenly across the GPU cluster**. For example, using 8-way sequence parallelism with the Wan2.1 14B model reduces the additional memory requirement to under 9 GB per GPU.
>
> **Computational Overhead**.The core operations during the cache step are element-wise, making them memory-bound. Execution time can therefore be estimated based on the GPU’s memory bandwidth.
> For an NVIDIA H800 with a memory bandwidth of approximately 3.3 TB/s, the estimated time per cache step for the Wan2.1 14B model is around 0.05 seconds.
>
> **Communication Overhead**. The communication overhead is negligible. At the end of each cache step, synchronization is only required for a small scalar statistic computed per device. With Pdevices, the aggregated data size for communication is merely about $2 \times P$ Bytes per step, resulting in a minimal communication cost.
>
>
> We have added the analysis of memory, communication, and additional computational overhead to Appendix G.
>
> > W3: Do not discuss the potential impact on devices with limited memory.
>
> This is a thought-provoking question. We are also further exploring feature caching and inference solutions in industrial scenarios. Here we present a computation-loading overlap scheme for feature caching under limited GPU memory.
>
> We are considering performing the element-wise operations of the cache on the CPU side. This approach reduces data transfer overhead — only the memory of the last block needs to be transferred to the GPU. Meanwhile, the full GPU computation and the transfer of intermediate activations to the CPU can be overlapped.
>
> For element-wise matrix operations, GPU computation may be 5–10× faster than CPU (limited by memory bandwidth: ~200 GB/s for a CPU server vs. ~10008 GB/s for an RTX 4090).
>
> Taking the RTX 4090 as an example, the additional GPU memory overhead required by this scheme is only 1/layer_num of the current solution. For Wan2.1 1.3B with layer_num = 30, this means only the output of the last layer (1.5 GB) needs to reside on the GPU, while the rest of the cached data (43.5 GB) can be stored on the CPU.
>
> However, this approach introduces an additional overhead of about 4× per caching step (around 0.18 s per step). As a result, video generation will incur an extra ~9 seconds of overall computation time. But given that single-card video generation currently takes around 253 seconds, this added overhead is relatively minor.
>
> ---
>
> Finally, we sincerely appreciate the reviewer’s careful review and the critical questions raised — particularly regarding **the cross-task stability of $\alpha$**, **the analysis of additional GPU memory overhead**, and **the exploration under limited GPU memory scenarios**. These insights have been invaluable in helping us further improve the paper. We hope the above responses adequately address your concerns.

---

> > ### Comment · Reviewer_PyFd · 2025-11-28
> >
> > Thank you for the detailed response and for addressing my concerns. The authors have resolved the issues I raised in my initial review. Consequently, I have raised my score.

---

### Official Review · Reviewer_mevD · 2025-10-31

**Soundness:** 2
**Presentation:** 2
**Contribution:** 2
**Rating:** 6
**Confidence:** 2

**Summary:**

This paper presents ScalingCache, a training-free method to accelerate Diffusion Transformers (DiTs). It improves upon standard feature caching by introducing a pre-computed scaling factor (alpha) for more accurate feature prediction and a dynamic caching strategy to adaptively skip computation steps. The method achieves significant speedups (2.5-3.1x) with minimal quality loss on major text-to-video and text-to-image models.

**Strengths:**

1. Effective: The differential scaling with alpha is a lightweight approach to improve feature prediction accuracy.
2. Strong Results: The method delivers impressive speedups while preserving high visual fidelity, outperforming prior methods in key metrics.
3. Practical: As a training-free solution, it is easy to apply to existing models without expensive retraining.

**Weaknesses:**

1. Robustness of Alpha: Is calculating the alpha coefficient from only 50 prompts sufficient for generalization across diverse inputs? The paper should discuss the method's robustness and show potential failure cases.
2. Analysis of Dynamic Caching: The ablation study confirms the dynamic caching strategy is useful, but lacks a deeper analysis. How does it adaptively change intervals for different content (e.g., static vs. dynamic scenes)?
3. VBench Score Breakdown: The analysis of VBench scores is too general. A breakdown by dimension (e.g., image quality, temporal consistency) is needed to clarify where the method truly excels. An explanation for why it doesn't achieve top scores on all models would also be helpful.

**Questions:**

Please see the weakness.

---

> ### Author Response · Authors · 2025-11-21
>
> We sincerely thank the reviewer for their careful review and recognition of our work.
>
> > W1: Robustness of Alpha.
>
> As shown in Figure 6 of the paper, the mean alpha demonstrates convergence as the number of prompts increases. Similarly, Reviewer 6bkW raised the same concern, and Reviewers PyFd and BDzQ suggested further analyzing the robustness of alpha across tasks. Therefore, we conducted additional experiments to validate the results across different subtasks (each subtask consisting of 10 prompts, with each prompt generated using 5 random seeds).
>
> For each individual sub-task $i$, we computed the corresponding metric $\alpha_i$. Additionally, we aggregated results across all samples​ to derive a global metric $\bar \alpha$. By analyzing the discrepancy between $\alpha_i$ and $\bar \alpha$, we quantitatively assessed the degree of deviation​ of each sub-task from the overall performance. This approach allows us to systematically evaluate task-specific variations and validate the generalizability of $\alpha$.
>
> | sub-task category      | $\lvert \alpha_i − \bar \alpha \rvert $ |
> |------------------------|----------------|
> | Motion-Rationality     | 0.008       |
> | Composition            | 0.007       |
> | Human-Interaction      | 0.009       |
> | Material               | 0.021       |
> | Mechanics              | 0.014       |
> | Dynamic                | 0.011       |
> | Static                 | 0.017       |
> | Random                 | **0.006**       |
>
> We have added a discussion on this aspect in Section 4.4 of the paper. As shown in Table above, the mean value of the global $\bar \alpha$ is 0.58. We found that the alpha values calculated from different majority subtasks deviate by no more than 2.5%, indicating that $\alpha$ exhibits good cross-task stability.
>
> If you are interested, you may refer to our response to Reviewer BDzQ, where we have also included an analysis of cases where the alpha deviates significantly for specific samples.
>
> > W2: Analysis of Dynamic Caching.
>
> Thanks for the feedback. We have further elaborated on this point in the paper to improve clarity for the readers. The motivation for setting the dynamic $\delta_s$ is illustrated in Figure 5 of the paper.
> As shown in Figure 5 of the paper, we observe that under the same LPIPS-based video quality, dynamic scene prompts require a smaller $\delta_s$, while static scene prompts require a larger $\delta_s$. The $\delta_s$ estimated from the first $S_f$ frames aligns well with this observation. As a result, we can achieve overall performance gains.
>
> Considering the increase in the number of high-quality generated videos, we consider videos with LPIPS < 0.1 to be of relatively good quality. Therefore, our goal is to achieve a higher percentage of videos with LPIPS < 0.1 under the same acceleration ratio as shown in the Table below.
>
> | Model           | SpeedUp | LPIPS  | dynamic $\delta_s$ | fixed $\delta_s$ |
> |----------------|---------|--------|-----------------------------|----------------------------|
> | Wan2.1 1.3B    | 2.5$\times$    | 0.071  | 82.68%   | 79.23%   |
> | Wan2.1 14B     | 2.5$\times$   | 0.083  | 69.49%   | 66.38%   |
> | HunyuanVideo   | 2.3$\times$    | 0.049  | 97.17% | 94.89%   |
>
> > W3: VBench Score Breakdown: The analysis of VBench scores is too general.
>
> Thank you for your suggestion. Adding sub-metrics is very helpful for result analysis.
>
> We reported the metrics provided in the MixCache[1] paper, where the Vbench scores are all relatively close to the original score of 84.05 — specifically, 83.97 (MixCache) and 83.87 (ScalingCache). Since MixCache has not been open-sourced, **we regret that we are unable to conduct further analysis.**
>
> We have provided the variation chart of Vbench sub-metrics for the HunyuanVideo model under high acceleration ratios in Section 4.3 of the paper. We added a comparison with the contemporary open-source work EasyCache[2], evaluating on the sub-metrics of Vbench as shown in Figure 13 of the paper. We found that our advantage in Vbench scores largely stems from the "dynamic degree" dimension, demonstrating our method's strength in handling dynamic scene changes.
>
> [1] Wei, Yuanxin, et al. "MixCache: Mixture-of-Cache for Video Diffusion Transformer Acceleration." arXiv preprint arXiv:2508.12691 (2025).
>
> [2] Zhou X, Liang D, Chen K, et al. Less is Enough: Training-Free Video Diffusion Acceleration via Runtime-Adaptive Caching[J]. arXiv preprint arXiv:2507.02860, 2025.
>
> ---
>
> Finally, we sincerely appreciate your recognition of our work, and we hope the above analysis can address your concerns.

---

### Author Response · Authors · 2025-11-21
**Summary of Rebuttal and Manuscript Revisions**

Dear Reviewers, Area Chairs, and Program Chairs,

We sincerely thank all four reviewers for their constructive comments and insightful auestions, which helped us refine our work.

*We would like to express our sincere gratitude to the reviewers for their appreciation of our work.*

- Reviewer mevD​ commended our work as "**easy to apply**"​ and described it as "**a lightweight approach.​**"

- Reviewer PyFd​ highlighted its "**sophisticated technical design**,"​ emphasized its **engineering friendliness**, and praised the excellent experimental results.

- Reviewer 6bkW​ noted that our manuscript features "**well-defined formulations, clear structure, and fluent exposition.**"​

- Reviewer BDzQ​ specifically appreciated the **rigorous derivation process, clear algorithmic steps, and highly competitive experimental outcomes.**

During the response period, we carefully try our best to provide feedback and conduct supplementary experiments to all comments from reviewers, **here are the modifications we have made to the manuscript**.

- Expand the discussion on feature cache strategies for the U-Net architecture in the related work section.
- Correct the symbol errors in some formulas within Section 3.2.
- Add an analysis of the additional computational, communication, and memory overhead introduced by ScalingCache in Appendix G.
- Further elaborate on the motivation for using dynamic error thresholds in Section 3.3.
- Include a radar chart of Vbench sub-metrics under different acceleration ratios in Section 4.3.
- Introduce a new Section 4.4 to analyze the cross-task stability of the alpha parameter.
- Add Figure 13 in the appendix, comparing Vbench sub-metrics with prior work to highlight the key advantages of ScalingCache.

*We have uploaded the revised version of the paper, with modified sections and text highlighted in blue.*

---

### Meta-Review · Area_Chair_5KcR · 2026-01-09

**Summary:**

This paper introduces ScalingCache, a training-free inference acceleration framework specifically designed for Diffusion Transformers (DiTs) in image and video generation. The authors propose a dynamic caching mechanism that leverages temporal redundancy in hidden states by using a precomputed differential scaling factor ($\alpha$) to reuse previous activations. By combining zero-order and first-order predictions, ScalingCache adaptively skips computation steps during the denoising process, achieving significant speedups (2.5× to 3.1×) across various state-of-the-art models like Wan2.1, HunyuanVideo, and FLUX while maintaining high visual fidelity.

Strengths:

- The paper addresses a highly practical and important bottleneck—the high computational cost of DiT inference—without requiring expensive retraining or fine-tuning.

- The technical design is sophisticated, introducing a differential scaling factor that improves upon simple feature caching and effectively handles prediction errors.

- The experiments are comprehensive, covering multiple SOTA models and both image and video modalities with a wide range of metrics (VBench, LPIPS, CLIP Score).

- The method is engineering-friendly, requiring minimal hyperparameter tuning and no complex scheduling logic for integration.

Weaknesses:

- The generalization of the $\alpha$ coefficient is a concern, as it is estimated offline using a small set of prompts; its performance on out-of-distribution or rare prompts is not fully explored.

- Certain edge cases, such as "static-to-dynamic" video transitions or high-speed movement, remain challenging and could lead to quality degradation like video flicker.

- The claim of "no additional overhead" is slightly overstated, as storing cache tensors and calculating dynamic errors involves non-negligible memory and communication costs in parallel settings.

Overall, the paper presents a well-motivated and technically sound framework for DiT acceleration. While some concerns regarding the robustness of offline estimation and missing baselines exist, the consistent speedup and near-lossless quality across major models make it a valuable contribution to the field.

**Reviewer Concerns:**

One reviewer actively engaged in the author-reviewer discussion and indicated a willingness to raise their score. Following this, the Area Chair (AC) thoroughly examined all responses. The authors adequately addressed the raised concerns by providing comprehensive clarifications and experimental results

**Reviewer Scores:**

After addressing most concerns in the rebuttal, one score is likely to turn positive (Reviewer PyFd: 4 $\rightarrow$ 6), resulting in three positive scores total (Reviewers mevD, PyFd, and 6bkW at 6, with Reviewer BDzQ remaining at 4).

---

### Decision · Program_Chairs · 2026-01-26

Accept (Poster)